# Natural Swimming Ponds as an Application of Treatment Wetlands—A Review

Wojciech Walczak, Artur Serafin * and Tadeusz Siwiec

Department of Environmental Engineering and Geodesy, University of Life Sciences in Lublin, Leszczyńskiego 7, 20-069 Lublin, Poland; wojtek.walczak@up.lublin.pl (W.W.); tadeusz.siwiec@up.lublin.pl (T.S.)
* Correspondence: artur.serafin@up.lublin.pl

**Abstract:** Natural swimming ponds using treatment wetlands (TWs) as an element of treatment of swimming water are an ecologically beneficial alternative to conventional pools. Unlike conventional swimming pools, in natural swimming ponds, the water treatment avoids the use of chemical methods and is based on the phenomenon of water self-purification and the rhizofiltration capacity of repository macrophytes in TWs of the regeneration zone, as well as on typical physical filtering processes (e.g., straining, sedimentation, or flotation), physicochemical filtration (physical and chemical adsorption, mainly of phosphorus), and biological filtration (nitrification and denitrification). Market solutions usually arise from the implementation of water treatment solutions used in small garden ponds, which are not typical for ponds; moreover, they are expensive and difficult to use and maintain. Therefore, they require the development of a dedicated system that improves the functioning and usability of the filtration system. A modular, compact filtration system for the treatment of water by physical and biological methods, made of polymer composites and with replaceable filtration modules and essential equipment (e.g., skimmers, dispensers, and filtration mats), is a solution expected by many pond users. It enables the exploitation of the natural functions of ponds and contributes to the preservation of biodiversity and active recreation in a biologically living aquatic ecosystem.

**Keywords:** swimming ponds; wetland system; nature pool; filtration chamber; mineral filtration





## 1. Introduction

### 1.1. Specificity of Natural Swimming Ponds

Natural swimming ponds ('nature pools') are artificial water bodies separated from surface waters and groundwater, distinguished by the absence of chemical disinfection or a water sterilization system. They are treated by means of biological processes, using macrophytes—plants rooted in the bottom of the pond in its wetland system—and filters with a mineral bed, in which the processes taking place are mainly physical [1–4].

Natural swimming ponds are, therefore, living systems in which the same processes take place as in natural water bodies. Wetland systems support and control these processes to varying degrees and function in completely different conditions than in wastewater treatment plants. They usually operate only during the growing season (i.e., the swimming season). The water is treated continuously in a closed process and is not released into the environment. The concentrations of organic matter, especially nutrients, to be treated are very low (phosphorus measured in micrograms, not milligrams), as are the loads of pollutants, but the hydraulic load is high [5].

In the era of the greening of socioeconomic life, such solutions have been developing in both Central Europe and Mediterranean countries since the early 1980s. The first natural swimming ponds intended for recreation were created in Austria and Germany, and from there they spread to nearly all of Europe [3,6,7]. Most natural swimming ponds are small private systems for family use, with a small number of bathers [3], although similar solutions are increasingly encountered in much larger community or city bathing areas [5].

The increase in the number of nature pools is mainly attributed to health concerns regarding the chemicals used in typical swimming pools, in which the water has long been treated using strong oxidants with the capacity to oxidize admixtures or organic and inorganic pollutants present in the water, such as chlorine gas, chlorine dioxide, sodium hypochlorite, or ozone [8–10]. These substances cause irritation and skin allergies, as well as respiratory diseases [11]. People may also have health concerns regarding the physical water treatment techniques, e.g., the use of ultraviolet radiation, although these fears are not fully justified [12].

Other extremely important advantages are the aesthetic appearance of a natural swimming pool reminiscent of a small lake, the attractive spatial composition of the landscape, and the relatively low costs of constructing and maintaining the facility [3,13].

A challenge in the use of natural swimming ponds, however, is to maintain adequate water quality and the pond's utility and health assets. The literature on the subject lacks descriptions of suitable water treatment systems that would effectively eliminate pathogens of human and animal origins, as well as non-fecal pathogenic microbes [14–16].

Diseases associated with this type of pathogen can be avoided through appropriate disinfection procedures and optimally selected non-invasive water treatment systems. The lack of such solutions in natural swimming pools is a serious problem in terms of the level of health risk. It also prevents maintenance of the recreational features of the swimming pond to ensure the appropriate colour, turbidity, transparency, and smell.

*1.2. Characteristics of Water Quality in Natural Swimming Ponds*

Although for people the main purpose of both natural and conventional swimming pools is recreational bathing, from microbiological, sanitary, chemical, and administrative perspectives, these are completely different systems [3].

In Poland, the parameters of swimming pool water should generally correspond to the quality of water supplied to households according to the acts on collective water supply and collective wastewater disposal [17]. These requirements are specified in the regulations on the requirements that must be met by water in swimming pools (Regulation of the Minister of Health, 2015) [18], the guidelines of the Chief Sanitary Inspectorate on requirements pertaining to water quality and sanitary conditions in swimming pools (2014) [19], and the act on the safety of persons present in water areas (2011) [20]. These documents specify the acceptable contents of microbes and physicochemical parameters in the water, the frequency of water sampling together with the methodology and scope of the procedures and testing, and the course of action when irregularities are detected [21].

Despite the growing popularity of natural swimming ponds, in Poland no clear and coherent legal regulations on water quality in these ponds have been developed, nor are there any suggestions regarding the quality of circulating water. For this reason German standards (FFL 2017) [2] and Czech standards [22] are used. On this basis, examples of acceptable ranges of values for water quality parameters in swimming ponds are as follows: oxygen concentration: 4–12 mg/L; oxygen saturation: 80–120%; conductivity: 200–1000 μS/cm; pH: 6.9–9.0; calcium concentration: 30–50 mg/L; magnesium: 5–10 mg/L; ammonium ions: 0–4 mg/L; nitrates: 10–30 mg/L; nitrites: 0.0–0.001 mg/L; total phosphorus: 0.03 mg/L; sulphates: 0.40 mg/L. It should be noted that in swimming ponds using more technologically elaborate water filtration systems, the acceptable ranges for these parameters correspond to the lower values in the ranges presented above [23].

*1.3. Treatment Wetland Application in Natural Swimming Ponds*

Maintaining adequate water quality and utility parameters in swimming ponds requires the use of appropriate, preferably natural treatment methods.

The primary way to achieve this is using treatment wetlands (TWs), which function as natural purification technologies that effectively treat many different types of water. They are used all over the world and are growing increasingly popular because they require smaller capital expenditures than other water treatment solutions [5].

In the case of natural swimming ponds, TWs are associated with several types of solutions, which are often combined:

- Planted vertical flow filters with saturated media or freely drained;
- Unplanted vertical flow filters with saturated media or freely drained;
- Free water surface (FWS) wetlands with submerged or emergent vegetation;
- High-rate gravel or technical filters as elements of additional water treatment technology.

Planted vertical flow filters are wetlands planted with vegetation with a vertical water flow, functioning as a filtration bed for secondary or tertiary treatment of water or wastewater. The water is treated by a combination of biological and physical wetland processes. Following the initial treatment, the water is poured out or dispensed onto the surface from above via a mechanical system. It then flows vertically down through the filter matrix onto the bottom of the pond, where it collects in a drainage pipe. This system can operate in conditions with a high water level or as a freely drained system.

Unplanted vertical flow filters operate in similar conditions as above but without planted vegetation, and exploit the adsorption potential of the multilayered substrate itself.

FWS wetlands are wetlands with a slow water surface exposed to the atmosphere. Most natural wetlands are FWS systems, including peatlands (where the primary vegetation is mosses) and swamps, with emergent or submerged vegetation. Their purpose is to recreate the naturally occurring processes of a natural wetland area. When water slowly flows through the wetland, particles settle, pathogens are destroyed, and animals and plants utilize nutrients. This type of artificial wetland is widely used for advanced water treatments following a secondary or tertiary treatment of water or wastewater. It should be noted that planted vertical flow filters should receive smaller loads than other types of TWs for swimming ponds because the root zone can reduce the volume of empty spaces in the filter, thereby reducing the hydraulic conductivity of the system.

High-performance gravel filters or technical filters, used especially to eliminate phosphorus, can have even higher load factors, but they are not effective at removing pathogens [5].

An important factor in selecting and combining various TW options for water treatment is their specific efficiency at eliminating fecal bacteria (e.g., *Escherichia coli*) and excess phosphorus at an optimum water flow rate [5].

An effective water treatment based on the efficiency of TWs entails the modification of solutions adapted to the needs of the users of natural swimming ponds. Efforts are made here to eliminate biogenic substances by exploiting competition for nutrient compounds between algae, which negatively affect the organoleptic characteristics of water, and the swamp plants of wetland systems. The use of specially chosen minerals binding biogenic substances also exploits sorption phenomena taking place in the bed of the plant filter, as well as bacterial cultures. This limits the sanitary risk resulting from the use of this type of pond [24].

The solutions applied, however, are not compact and modular and are difficult to assemble and use, especially if there is a need for modernization. Sometimes an undesirable chemical treatment is used, which can affect fauna and flora [25], or UV lamps are utilized, which destroy as much as 90% of microbes, including useful ones playing an important role in improving water quality [26].

The development and implementation of a polymer, compact, and modular filtration chamber that is easy to assemble, use, and modernize for use in natural swimming ponds with TW solutions—in accordance with conceptual assumptions—will make it possible to improve the filtration system and regulate its efficiency without the need for specialist expertise. Of course the starting point will be the design of an entire filtration system for the needs and uses of a natural swimming pond, its size, and its functional structure.

## 2. Structure and Function of a Natural Swimming Pond as a TW Application

The simplest construction of a natural swimming pond is shown in Figure 1.

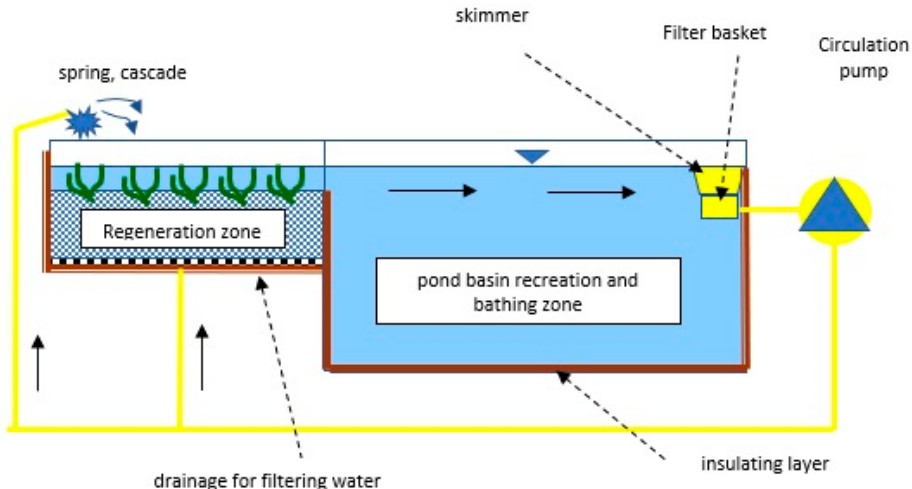

**Figure 1.** Major components of a natural swimming pond.

The simplest model of a natural swimming pond consists of a pond basin filled with water, lined with highly durable hydro-insulating polyvinyl chloride (PCV), a geomembrane (EPDM), or other impermeable materials (clay or bentonite mats). The use of an impermeable liner is meant to hold water and reduce contamination originating in the soil [7]. The finish of the basin varies, comprising stone slabs, 10–15 cm of fine and coarse gravel, stones, or even swimming pool tiles, depending on the design.

In functional terms, a natural swimming pond is divided into two zones: the shallow regeneration zone, functioning as a treatment wetland (TW), planted with aquatic vegetation, which helps to heat the water quickly and to successfully filter the water and sediment; and the deep recreation and bathing zone, which is intended for use [4,27,28].

The regeneration zone consists of a substrate composed of appropriate minerals (the filter), with a drain in the lower part that distributes the filtered water and appropriately selected plants growing in the upper part. The entirety is filled with water, which shares its surface with the bathing part of the pond. The circulation pump and filter are protected by a surface skimmer with a sieve that collects floating impurities, from which the pump draws water and pumps it to the drain under the filter. Some of this water can be pumped sideways over the filter where it sprinkles it, providing additional oxygenation. The water flows upwards from the drain to the surface and flows on to the bathing zone. The filter bed layer with plants functions as a plant filter, which traps organic biogenic substances and makes it a TW solution, as a planted vertical flow filter and FWS wetland.

The efficiency of plants in this filtration process is directly proportional to their biomass (ability to accumulate biogenic elements) and phenology (period of immobilization of biogenic elements). The plant biomass accumulates in the edge zone and can be removed by mowing, although the accumulation of plant biomass in the humification process in the form of fen peat has no negative effect on water quality [24]. The phytosanitary role of the regeneration zone is very important as well. The removal of microbial pathogens is based here on complex mechanisms—physical (sedimentation, adsorption, and filtration), chemical (oxidation, UV radiation, exposure to plant biocides, unfavourable water chemistry, or adsorption to organic matter and biofilm), and biological (predation, biological decomposition, antibiosis, or natural death)—often acting in combination [5,29,30]. The effectiveness of these mechanisms, therefore, depends on the synergistic effect of natural (environmental) and technical (design and operational) features, which act on different pathogenic microbes in different ways [31,32]. Similarly, TWs in the regeneration zone, depending on their construction, may take part in the removal of micropollutants, including substances originating in synthetic products and human activity, which are present in very

low concentrations in the environment (parts per billion (ppb) and lower). They include pharmaceuticals and personal care products (PPCPs), pesticides, industrial chemicals, endocrine-disrupting chemicals (EDCs) (including hormones), and nanomaterials [5,33,34]. The mechanisms involved in the removal of micropollutants from an inflowing TW include microbial transformation, uptake, and metabolism by plants; adsorption onto biofilms or the substrate; volatilization; abiotic degradation, including hydrolysis or photocatalytic oxidation; and other advanced reduction or oxidation reactions [35].

It is worth noting that macrophytes of the regeneration zone, depending on their species composition and life form, are also able to accumulate heavy metals, forming a biogeochemical barrier [36]. This is particularly important when filtration of pond water involves the use of components containing copper or silver (electrodes or UV filters), which can lead to the biomagnification of these elements in the food webs of the pond [37] and pose a long-term risk for the effectiveness of TWs [38]. Plant and animal periphyton communities and bacterial biofilms are also extremely important in this zone. They grow on plants themselves as well as the porous substrate they are found in (lava, sand, or gravel) and supplement filtration in this zone, improving its efficiency, particularly in the removal of the phosphorus fraction [39].

The bathing and recreation zone is much deeper than the regeneration zone, and its depth varies. Bottom sediment accumulates here and must be removed regularly. Sometimes, if the slope of the bottom is adequate and bottom outflow is used, the sediment can be transported to the sewage system or a sediment well [4,40].

The bathing zone and TW (regeneration) zone can be separated, either completely, using wood or stone, or through a succession of changes in the ground level. These zones can also be located in completely different parts of the recreational area, in accordance with the assumptions of the landscape architect [41].

The most important assumption for the functioning of this type of pond is the complete abandonment of the use of the chemicals used in many places for water treatment. This means that the water treatment is based on self-purification of the water. This is a positive physical and biochemical phenomenon of varying intensity levels, involving the sedimentation of suspended solids and mineralization of organic compounds, followed by the uptake of the material by plants and its incorporation into their own structure [42–44]. Therefore, it is the basis of the functioning of TWs. In this process, the physical, chemical, and biological factors are interdependent. The physical factors include the shape of the pond basin, water density and temperature, water flow rate (usually technologically induced by a pump), and water flow turbulence (in the case of a pond combined with a waterfall). The chemical factors include the contents of $CO_2$ and $O_2$. The biological factors include plants and animals living in the pond, which take up organic pollutants, break them down into simple compounds, and secrete them in this form into the water [24,42,44].

Most of the mechanical treatment takes place in the porous layer of the substrate filling the regeneration zone, to which water from the pond is pumped from above by the circulation pump [45]. Processes such as straining, sedimentation, flotation, and filtration take place within the bed [46,47]. The end result of the filtration process, therefore, depends on the quality of the water entering the filter, the type of filter material, the grain size and height of the bed layer, the type of filter, and the flow rate through the bed. Biological water treatment technologies exploit the activity of living organisms in an artificially created environment within the biological beds of a filtration chamber (separate or part of a modular system) or accompany mechanical filtration in a water treatment system. The purification processes taking place here are identical to those in a natural water body, i.e., transformations of nitrogen compounds by nitrifying and denitrifying microorganisms on a biological bed [48].

A mixed community of microorganisms (nitrifying and denitrifying) with high purifying capacity is formed along the path of the water flow through the filtration material, forming a biofilm [49]. Organic compounds entering the biological membrane by diffusion are retained there and then mineralized by biochemical decomposition. Oxygen

access in the bed is ensured by a natural air draught through the structures of the filter media [50]. An element commonly used to stabilize the most important parameters of the water of natural ponds, which is crucial to the proper functioning of biological filters, is $CO_2$ dispensers. They monitor the pH of the water and keep it at an appropriate level, applying carbon dioxide as needed to raise the carbonate hardness (KH) of the water, which performs the functions of a buffer. The artificial enrichment of the water with $CO_2$ is aimed at increasing its free reserves, which also ensures stable conditions for the development of aquatic repository plants. In this case, unwanted algae are less likely to take over the pond [23].

Bacteria, fungi, protozoa, algae, macrophytes, and animals take part in the aerobic decomposition of organic matter, in accordance with the functionality of the food web of the pond [42], and form the basis of the functioning of TWs [5]. Bacteria can utilize organic matter in the form of dissolved matter—even at very small concentrations—directly in cellular respiration, or in the form of particulate matter, secreting exoenzymes that decompose particles of matter into more easily absorbed forms [51]. Fungi, as heterotrophic organisms, play a similar role to that of bacteria. They take part in the biological decomposition (aerobic and anaerobic) of organic matter of plant and animal origins. They introduce some elements (taken up from decomposition of dead organic matter) into circulation in nature [52]. Protozoa in the aquatic environment utilize biomass accumulated in bacteria and fungi and take up organic substances in colloid form, and their species composition and abundance are closely correlated to the abundance of bacteria [42,53]. Algae (flagellates, cyanobacteria, diatoms, Zygnematophyceae, and green algae) take up carbon compounds and organic nitrogen from dissolved organic compounds. They also transform fatty acids, amino acids, urea, and peptones. They produce various photosynthetic pigments, with algae containing chlorophyll and producing oxygen as a result of $CO_2$ assimilation playing a particularly important role. They are food for many animals, protozoa, rotifers, molluscs, flatworms, roundworms, and larvae of aquatic insects. Their metabolism often involves the secretion of strong toxins that synergize in the water. This is particularly important in the case of a sharp increase in the biomass of algae making up the phytoplankton, causing algal bloom—a dangerous phenomenon often resulting in the intoxication of water bodies [24,42,54]. It is determined by high contents of biogenic substances, i.e., phosphorus and nitrogen fractions, causing excessive eutrophication of the water body [36,42,55].

Higher aquatic plants, both emergent and submerged (e.g., water hyacinth, common reed, rushes, or irises), function as filters. They take up vast amounts of mineral salts from the water. Of particular importance are repository plants, i.e., those that retain large amounts of nutrients in their tissues, especially phosphorus. In this way they compete with phytoplankton and filamentous algae, which makes them a factor stabilizing the ecosystem of the natural swimming pond. Well-developed populations of watermilfoil and pondweed species limit the occurrence of algal bloom threatening the ecosystem of natural ponds [5]. The intensity of the water purification process is also influenced by the effect of the rhizosphere used to remove or detoxify contaminants (rhizodegradation). This is a set of phenomena occurring in the root zone of macrophytes, manifested as increased quantity and activity of soil organisms, which facilitates the transfer of nutrients from the soil to plants.

Nutrient uptake by plants is determined by biophysicochemical processes taking place between plant root systems and the soil. These involve the secretion of carbon dioxide and organic acids by the roots, and also the supply of dying tissues, which increases the solubility of various soil substances and results in the development of soil microbes [24]. Aquatic plants additionally help to maintain oxygenating conditions around their root zones; this is particularly important on flooded filters, which are permanently filled with water [5]. Bacteria living in the vicinity of roots play the main role in the decomposition of organic compounds, and plant root secretions influence the species composition of the rhizosphere. Rhizofiltration as an element of phytoremediation is, therefore, one of the most effective means of water self-purification in the natural conditions of open water

bodies [24,56]. Plants on filters also shade and cool the water. Shade minimizes the spread of filamentous algae in shallow water, because filamentous algae compete with aquatic plants not only for nutrients but for light as well. Their metabolic activity associated with photosynthesis also leads to a temporary increase in the concentration of oxygen in a given area, which enables the aerobic decomposition of organic matter and limits the metabolic intoxication of the water body (anaerobic decomposition entails the formation of compounds harmful to ecosystems, such as indole, skatole, and mercaptans). Macrophytes also provide mechanical protection for shore areas, which prevents turbidity caused by sediment stirred up by water movement. They are also habitats for a large number of aquatic invertebrates and amphibians. Helophytes with strong root or rhizome growth, e.g., of the genus *Carex*, *Juncus*, *Schoenoplectus*, *Bolboschoenus*, or *Cyperus*, are often used in such zones [5]. Submerged macrophytes are also habitats for the development of many species of zooplankton (protozoa, rotifers, microscopic crustaceans, and insect larvae), which through trophic relationships act as biological regulators of the abundance of phytoplankton (algae and cyanobacteria). This relationship is exploited for the application of natural methods of aquatic ecosystem regulation based on biomanipulation (effect on the abundance of a specific element in the food chain).

In the case of natural swimming ponds, the aim of biological regulation is to find a quantitative biocoenotic balance between the phytoplankton and the zooplankton grazing on it based on the functional traits of food webs. It should be noted, however, that a greater abundance of zooplankton does not always cause a reduction in the amount of phytoplankton. This may be due to the various feeding characteristics of zooplankton, including their moderate consumption, which stimulates the development of phytoplankton; the selective grazing of edible forms of phytoplankton, allowing inedible forms to replicate; the metabolic secretion of phosphorus and nitrogen; or the accumulation of biogenic compounds by some forms of phytoplankton when they pass through the digestive tracts of consumers. The use of zooplankton to limit the amount of phytoplankton is, therefore, very difficult to regulate, and at the same time demonstrates the vast complexity of the various relationships in aquatic ecosystems [42,57].

The conditions of plant life in natural swimming ponds are very similar to those in natural sites. The choice of suitable plants is, therefore, based on the Ellenberg indicator, defining the preferences of individual plant species with regard to the most important environmental factors [58], determined by the design specifications of the pond. Therefore, it should be remembered that well-chosen, often native plant species in the regeneration zone based on TW solutions (Figure 2) help to keep the biophysicochemical parameters of the water at appropriate levels; eliminate or inactivate excess biogenic substances; limit the excessive development of harmful algae and cyanobacteria; destroy dangerous bacteria of the genera *Salmonella*, *Escherichia coli*, *Enterococcus*, and *Pseudomonas aeruginosa*; remove toxic compounds, nitrates, heavy metals, phenols, and cyanides from the water; protect the edges of the water body; and create optimal conditions for the development of aquatic fauna [59].

The phenomenon of water self-purification taking place in the regeneration zone of a pond with TW functions sometimes must be supported by the use of filtration systems complementing biological processes. The solutions applied here include cascades and fountains, water pumps, aerators, and carbon dioxide dispensers, or a combination of physical mechanical, mineral, and biological filtrations chambers, which maintain appropriate water quality in swimming ponds.

Therefore, bathing ponds function as quasi-natural systems based on natural processes linking the issues of the biotope and biocoenosis in the pond ecosystem. Therefore, there are no systemated methods to improve the efficiency of such a system. Ad hoc remedial actions are associated with the permanent monitoring of the biological state of the phytocenosis of the regeneration zone and the organoleptic and physicochemical parameters of the bathing water. In the absence of water treatment effects, it is recommended to replace plant species, change the proportions of individual subzones, and supplement natural water treatment

methods with various technological solutions (filtration). The effectiveness of their operation is determined on the basis of the quality indices of the treated water before and after the filtration process, and on the basis of the rate of growth of bed pressure losses and the bed's absorptive capacity. The entire biotechnological system of the water purification process is, therefore, adjusted in terms of efficiency to the needs of the investment and is subject to ongoing control, maintenance, and modification.

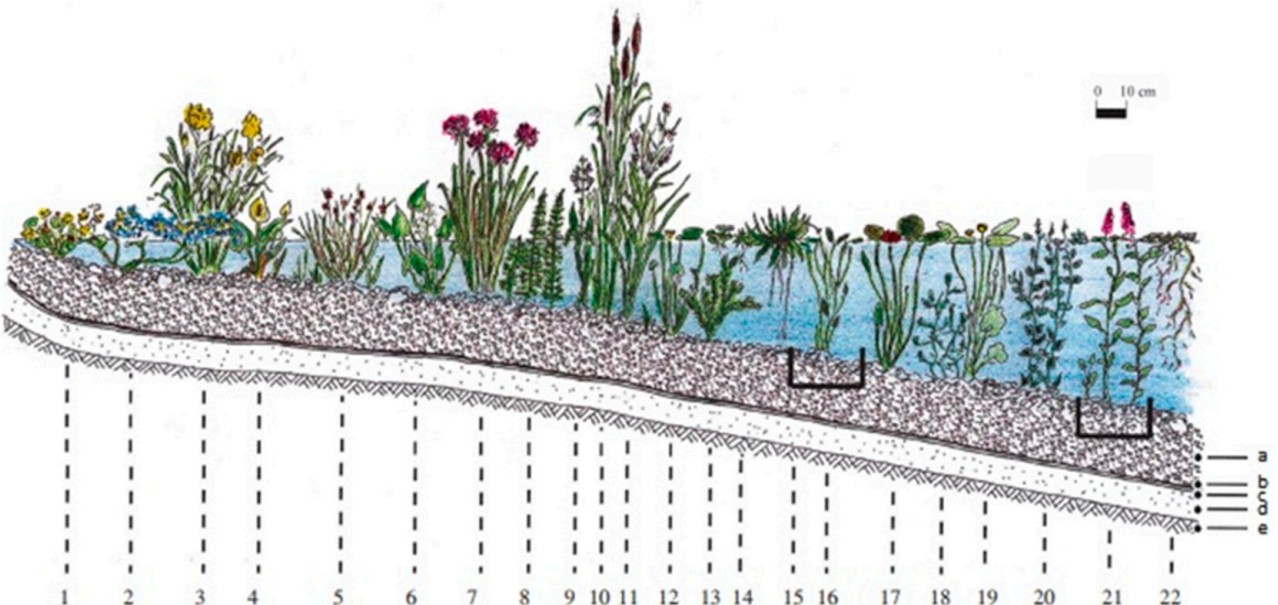

**Figure 2.** Regeneration zone of a swimming pond with examples of macrophytes ([24], modified): 1—*Caltha palustris*; 2—*Myosotis palustris*; 3—*Iris pseudoacorus*; 4—*Calla palustris*; 5—*Juncus* sp.; 6—*Alisma plantago—aquatica*; 7—*Butomus umbellatus*; 8—*Hippuris vulgaris*; 9—*Sagittaria sagittifolia*; 10—*Typha angustifolia*; 11—*Sparganium erectum*; 12—*Nymphoides peltata*; 13—*Ceratophyllum demersum*; 14—*Hydrocharis morus-ranae*; 15—*Stratiotes aloides*; 16—*Aponogeton distachyos* *; 17—*Nymphaea alba*; 18—*Elodea canadensis*; 19—*Nuphar lutea*; 20—*Potamogeton natans*; 21—*Pontaderia cordata* *; 22—*Trapa natans*. Note: * alien species requiring storage in winter (other species are native); a—gravel; b—PCV film; c—geotextile, d—sand, e—natural ground.

## 3. Modifications of Solutions for the Construction and Operation of Swimming Ponds

There are several criteria for classifying quasi-natural swimming ponds. In terms of the intensity of their use for bathing, the following types are distinguished:

- Extensive ponds, usually with 30% of the area designated for free swimming and 70% as a plant (regeneration) zone. Water circulation between these zones takes place owing to natural physical processes, namely different wind strengths and differences in temperature between the warmed shallow zones and the deeper, colder part of the pond. Ponds of this type are fully reminiscent of a natural environment;

- Intensive ponds, where the regeneration zone can be much smaller in relation to the recreation zone, as water circulation is forced by a pump and sometimes by more complicated water filtration systems. These include the use of various types of filters in various functional combinations, in some cases with additional equipment [13].

On the basis of the type of water treatment technology used, five types of natural swimming ponds are distinguished.

Type 1—Technology-free ponds, the simplest type, with no additional technology, natural water circulation, a recommended size of at least 120 m$^2$, and a minimum 7:3 ratio of regeneration zone to recreation zone (Figure 3).

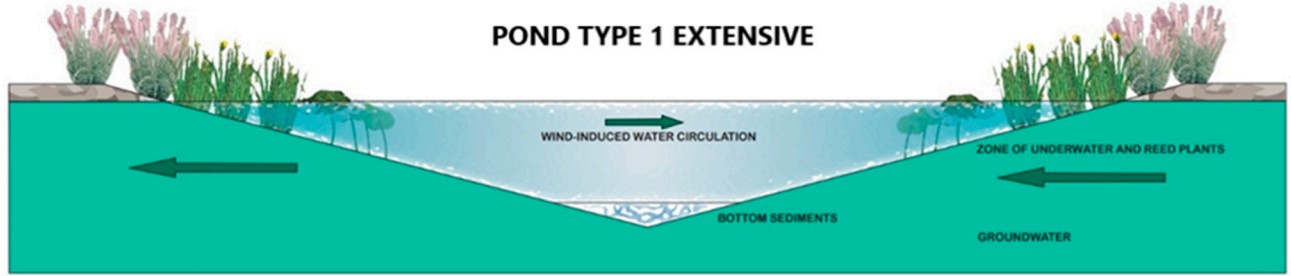

**Figure 3.** Diagram of the operation of an extensive swimming pond (type 1).

Type 2—Hydro-botanical ponds, without filter beds. In this case the water flow is forced by circulation pumps, which draw water from one part of the pond using a skimmer and pump it to the opposite part, where a shallow layer of water is overgrown with plants. In addition, a pipeline is installed for the suction of water from the lowest point of the bottom of the recreation zone to remove sediment from the bottom. The recommended size for these ponds is at least 120 m², with a minimum 1:1 ratio of the regeneration zone to bathing zone (Figures 4 and 5).

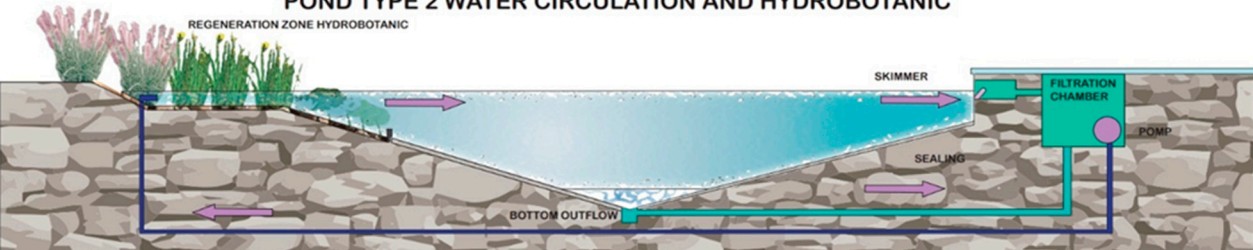

**Figure 4.** Diagram of the operation of a hydro-botanical swimming pond (type 2).

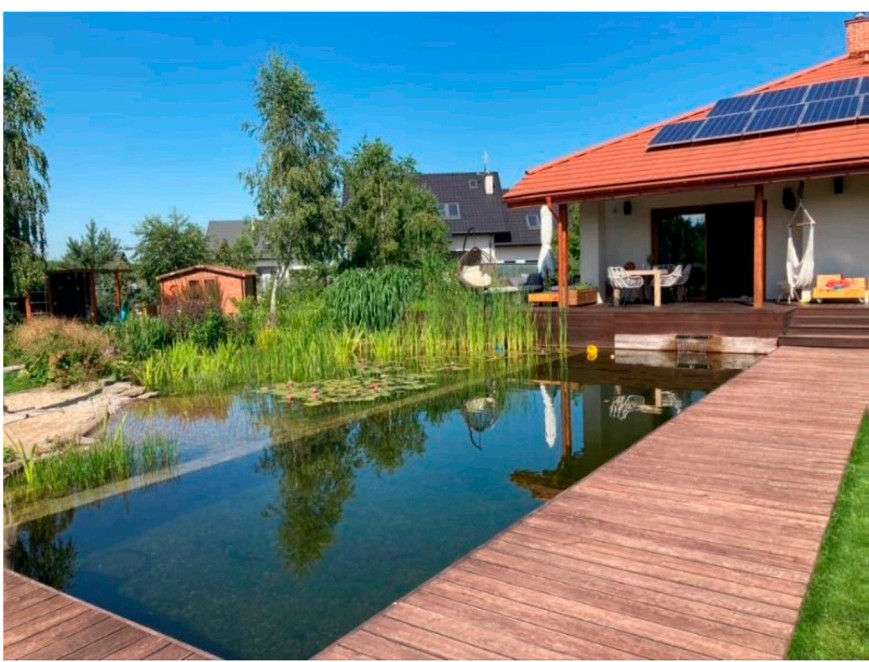

**Figure 5.** A hydro-botanical swimming pond (photo by W. Walczak).

Type 3—Hydro-botanical ponds with a filter filled with a mineral medium and other filtration systems. The pond is equipped like a type 2 pond with an additional filter filled with a mineral medium and planted plants. Owing to the use of filtration with a slow water

flow (3–5 $m^3/m^2$/day), the regeneration zone can be smaller. In this way, natural water self-purification processes are intensified, maintaining the high efficiency of the system in the long term. To maintain adequate water quality, additional devices are used as needed, such as pool robots, bottom drains, surface skimmers, or carbon dioxide dispensers. The area should be at least 80 $m^2$, with a minimum 2:3 ratio of the regeneration zone to bathing zone (Figure 6).

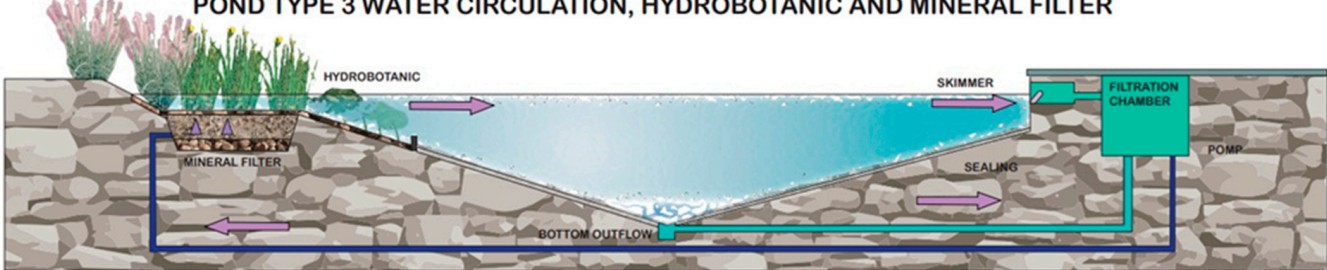

**Figure 6.** Diagram of the operation of a hydro-botanical swimming pond with a mineral filter (type 3).

Type 4—A pond type with rapid water flow in a filter with a mineral medium. These ponds can look similar to traditional pools, with a completely or partly separated regeneration part. They have a separate filter installed with a suitably chosen medium and a high flow rate (from 15 $m^3$/day). The role of the regeneration zone is still important, although less so than in the pond types described above. The conservation and care processes are often automated, and the recommended pond size is from 50 $m^2$ (Figures 7 and 8).

![Diagram labeled "POND TYPE 4 WATER CIRCULATION, HYDROBOTANIC AND DRY-SPRINKLE MINERAL FILTER" showing a cross-section of a swimming pond with dry-sprinkle mineral filter, hydrobotanic zone, skimmer, filtration chamber, pomp, sealing, and bottom outflow.]

**Figure 7.** Diagram of the operation of a hydro-botanical swimming pond with accelerated mineral filtration (type 4).

Type 5—A pond type with advanced technology, including combined water purification technologies. These are technologically complicated and require considerable financial outlays. The regeneration zone is entirely separate from the bathing zone and makes up only 25% of the area of the pond, or in some cases there is no regeneration zone at all. Therefore, plants do not play a major role in water filtration, but an increased number of other water treatment devices compensate for the absence of plants. The water is treated technologically using drum filters, modular filters, and bioreactors. This requires a large amount of energy, and the conservation and care processes are fully automated (Figure 9) [23].

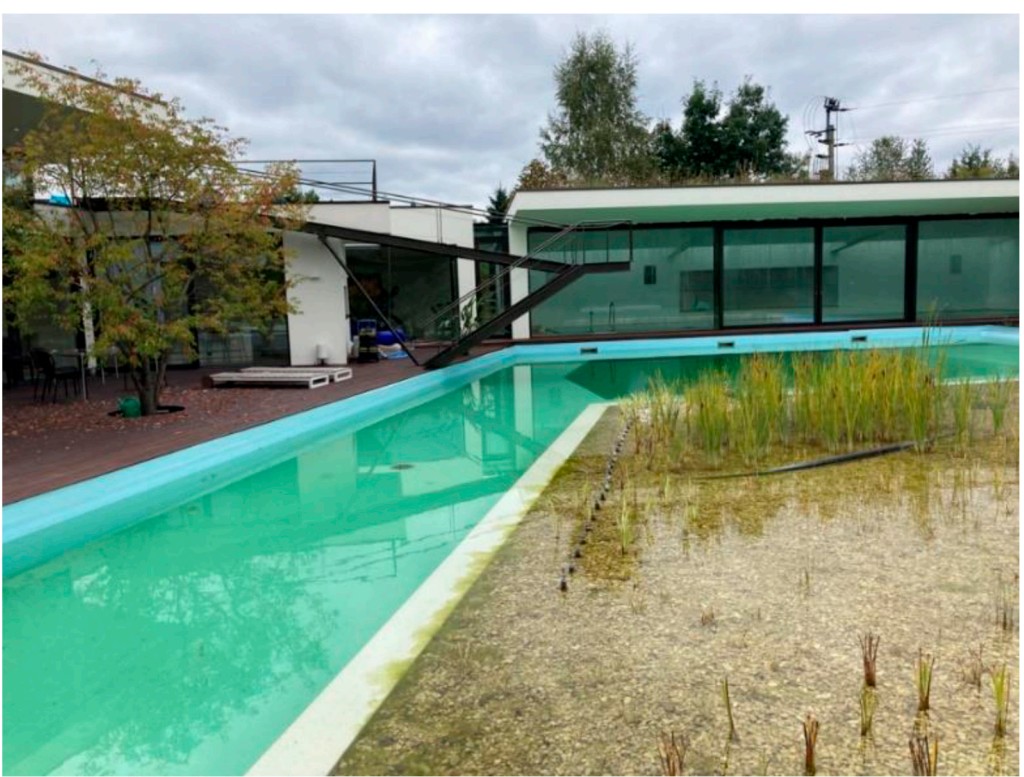

**Figure 8.** Swimming pond with a regeneration zone and accelerated mineral filtration (photo by W. Walczak).

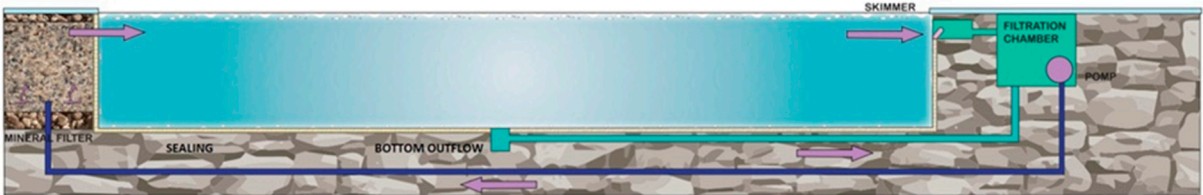

**Figure 9.** Diagram of the operation of a swimming pond with no regeneration zone (type 5).

## 4. Types of Filters

Filters improving water quality play an important role in type 2 and 5 ponds. These filters can have a variety of structures, media, and technological parameters. A crucial parameter is the water flow rate through the bed, and here we can distinguish typical slow biological filters with a flow rate of about 0.1 m/h and high-speed filters with flow rates ranging from 5 m/h to 20 m/h, in which physical processes are crucial and chemical processes have a minor role. Filters with low filtration speeds are usually open filters, in which the driving factor is the height of the water column above the bed, while filters with higher speeds are closed, with filtration forced by increased water and air pressure above the bed. An important parameter of filters is the type of medium, which may consist of inactive minerals (anthracite, sand, or chalcedony) or those entering into reactions with water components (dolomite, marble, or ionites) and sorption materials (activated carbon). The type of material is correlated with the grain size of the bed by the technological process. A coarser grain means a smaller water treatment effect, lower hydraulic resistance, and a higher filtration speed and water throughput, and vice versa.

Filters can function as single-block elements with one, two, or multiple chambers (with mechanical, chemical, and biological filtration); multi-block elements; or modular

elements, with the configuration and number of modules selected individually for each solution. They can be installed directly in the pond, e.g., under a deck or footbridge, or buried on the shore, directly or on a reinforced concrete screed.

Their efficiency can be increased by operating in systems with dedicated accessories (skimmers, circulation pumps, mats, sieves, filter sponges, UV lamps, $CO_2$ dispensers, pumps dispensing microbiological agents, bottom drains, sludge vacuums, bioballs, filter media such as Kaldnes media, etc.), using dedicated adsorption beds (selected natural adsorbents, artificially modified and high-performance minerals) and specific microbial preparations, depending on the inventiveness and suggestions of water technology companies.

Several basic types of filters are most common in practical solutions: mechanical filters (drum and sponge filters); filters driven by gravity and pressure; filters filled with various types of artificial or natural filtration materials; and various combinations of filters with auxiliary equipment.

In the case of mechanical drum filters, water from the pond reaches the center of the drum by means of gravity or pressure (Figure 10). From there it passes through the small holes in a microsieve to the filter chamber, leaving all of the impurities (physical particles, suspended solids, algae, dead organic matter) on the inside of the drum. When the microsieve is heavily soiled, the water levels of the filter chamber and the inside of the drum begin to differ. A water level sensor is activated, and the controller switches on the rotation of the drum and the sprinklers. The water under pressure washes the impurities off the inner side of the microsieve and into a special gutter, through which it is discharged into the sewage system. After one rotation of the drum, the microsieve is cleaned. When impurities again collect on the inner side of the microsieve, the cycle is repeated. The sprinklers can be fed with water from the water supply network or by a circulation pump in the system.

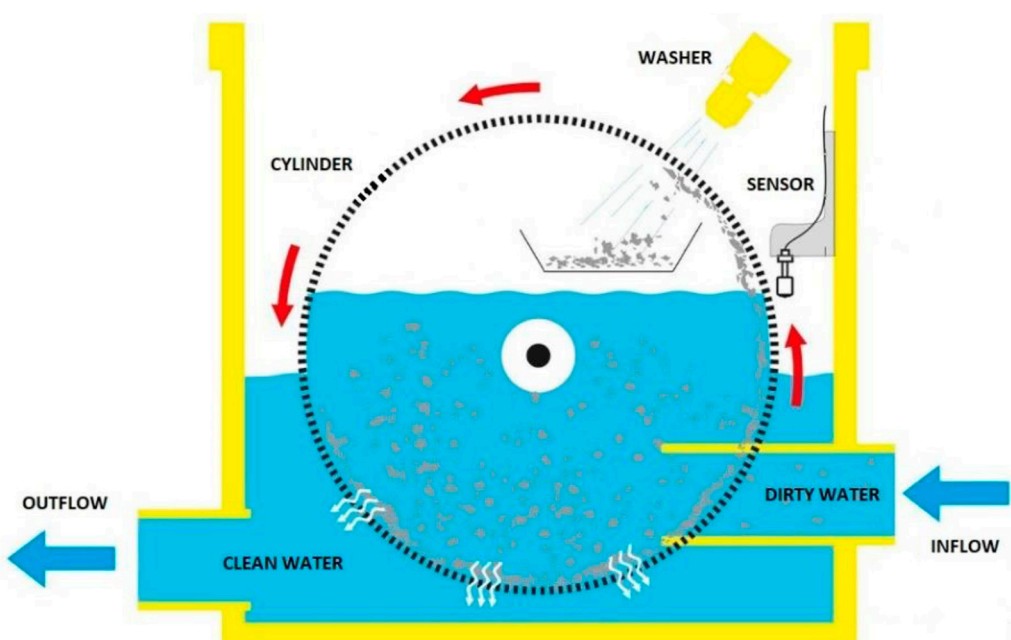

**Figure 10.** Diagram of the operation of a drum filter ([60], modified).

Drum filters are often used as elements of preliminary filtration, which only in combination with a highly efficient biological filter, forming a mechanical/biological single-block drum filter, can guarantee efficient water treatment. The biological filter in this system is filled with a porous material (bioballs, filter media such as Kaldnes media, geomembrane hydraulic fittings, etc.) with a biologically active area of 800 $m^2/m^3$, which enables colonization by nitrifying microbes, improving the efficiency of the water purification process (Figure 11).

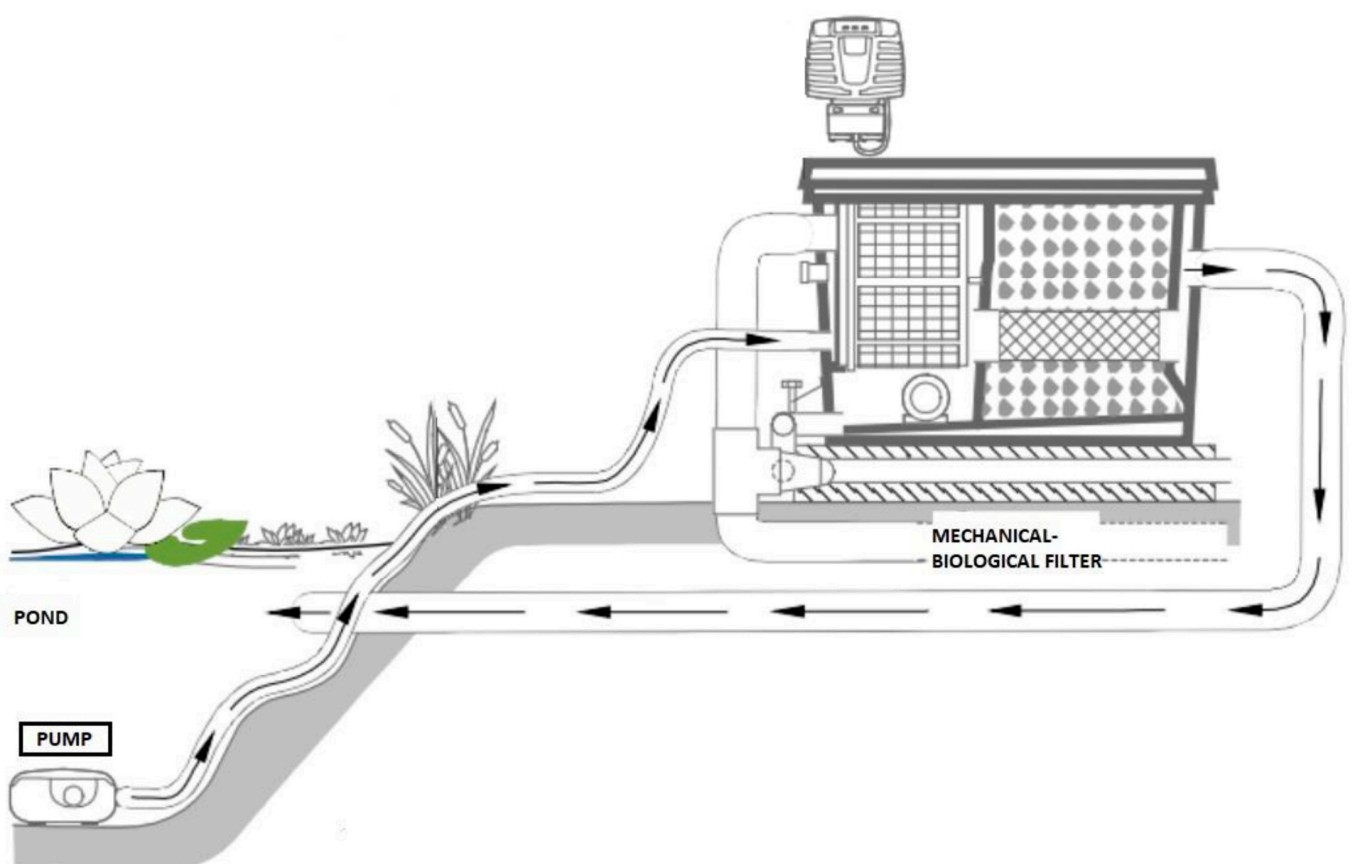

**Figure 11.** Diagram of the operation of a single-block mechanical–biological drum filter ([61], modified).

The rate of this type of water filtration depends on the size of the drum filter and the auxiliary technology. In market solutions, the efficiency of this type of system is guaranteed at flow rates of up to 20 m$^3$/h and pond volumes ranging from 120 to 300 m$^3$.

Mechanical sponge filters can have several chambers and are filled with filtering sponges of varying porosity levels (Figure 12), which ensures efficient mechanical filtration by means of the retention of physical contaminants and biological filtration through optimal colonization by various types of bacteria. Common additional elements in this type of filter include bioballs, porous biological cartridges, and Japanese filter mats made of polyester fiber and a waterproof binder, increasing the active area of the bed (Figure 13). The system is usually supported by a circulation or fountain pump with the power adjusted to the size of the filter and number of chambers. The functional efficiency of this type of filter depends on the size of the pond, its basic function, and the additional equipment used, and is guaranteed for a capacity of 90 m$^3$ at a flow rate of 12,000 L/h.

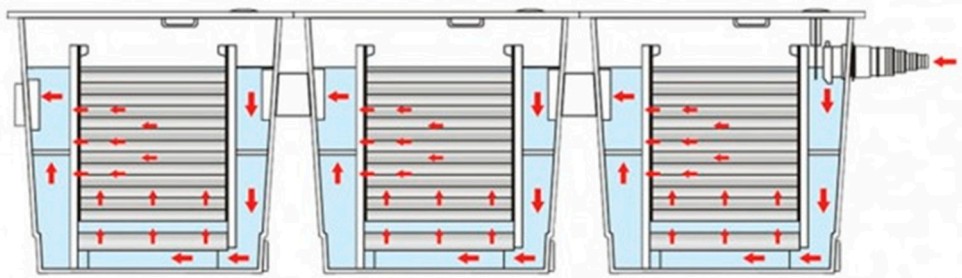

**Figure 12.** Diagram of the operation of a three-chamber sponge filter [62]. Red arrows indicate the water flow through porous filtration materials and between filter chambers.

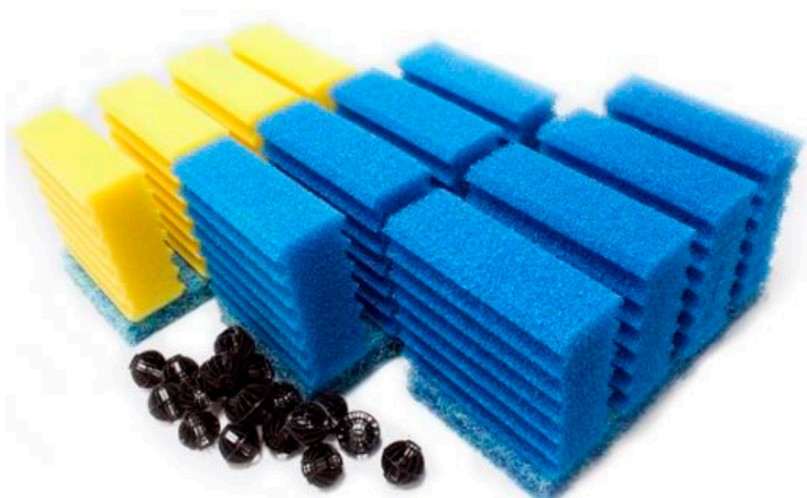

**Figure 13.** Sponges on a Japanese filter mat and bioballs used for biological filtration [63].

An interesting example of environmentally friendly solutions in water treatment technology in natural swimming ponds is a single-block filter with four sections. The first section is a settling chamber into which water flows through a skimmer. Next, the water flows through a filtration chamber filled with at least three filter mats, and then through a mineral filter with a bed consisting of adsorption material. The next chamber is a mechanical filter combined with a suction chamber (Figure 14) [64]. The suction chamber contains a pump that interestingly operates using suction rather than pressure. Then, the pump pumps water directly into the pond. The whole system is additionally equipped with a built-in carbon dioxide dispenser and a pump dispensing selected coagulants.

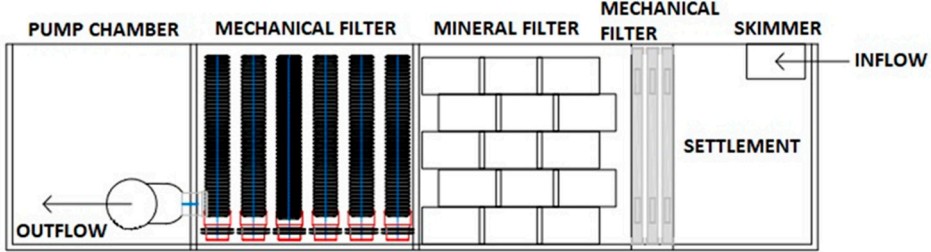

**Figure 14.** Diagram of the operation of a single-block filter with mechanical and mineral filtration and a suction section.

This type of solution is intended for type 3 swimming ponds, with a maximum capacity of 400 m$^3$. Owing to its innovative solutions, it does not kill the zooplankton that filter phytoplanktonic algae and cyanobacteria out of the water. A special system of suction baskets protects the zooplankton from being destroyed by the pumps and pumped into the mineral filtration zones. The filtration panels (baskets) of the suction chamber retain efficient filtering organisms, such as *Bosmina*, *Daphnia*, and *Cyclops*, measuring 1–2 mm in length, which due to their high density are able to filter the water in the chamber a dozen or more times a day. This solution results from the combination of the knowledge on biological, physical, and chemical processes associated with filtration during bathing water treatment with ecological phenomena of the pond ecosystem. This technology was invented by the landscape architect and ecologist Marcin Gąsiorowski [64] and is in common use in many European countries.

Larger natural swimming ponds (400–500 m$^3$) require slightly more elaborate water treatment systems involving the use of modular filtration systems adapted to their size (Figure 15).

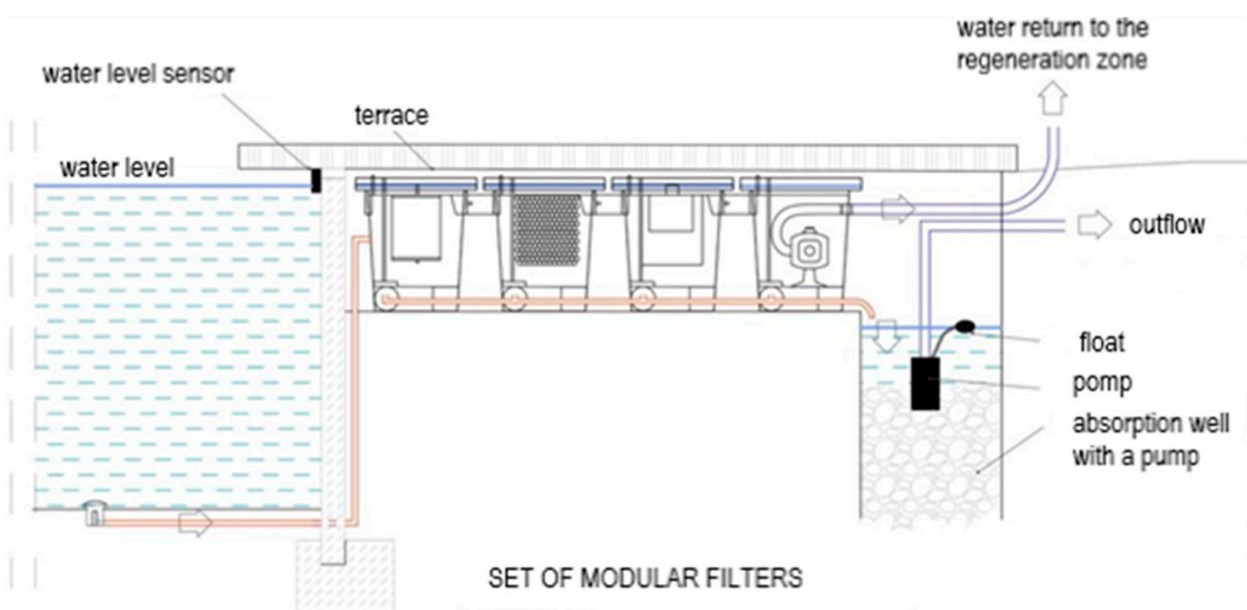

**Figure 15.** Diagram of the operation of a modular filtration system based on a mechanical drum filter and mineral and biological filter chambers ([23], modified).

The modules of mechanical, mineral, and biological filter chambers can be multiplied, depending on the water quality needs. Other commonly used elements include circulation pumps of sufficient power and extensive auxiliary equipment pieces (skimmers, aerators, automatic water filling systems, UV lamps, etc.) working with an intelligent controller with a self-purification function, so that the mechanical module is self-operating.

The typical market solutions, however, are difficult to install, utilize, and modernize, and require maintenance and repair from qualified water technology companies. They are usually not compact or modular. Sometimes filters are used with media that cause significant negative changes in the living conditions of fauna and flora or UV lamps that destroy useful microbes.

An innovative, dedicated, modular, compact, inexpensive, highly efficient water filtration system that is easy to install, operate, and regenerate based on a system of replaceable cartridges (mechanical, mineral and biological), which can be modified according to water quality requirements using biological methods, without the need for specialist expertise, is, therefore, a solution awaited by many users of natural swimming ponds. It will be an important element supporting the efficiency of TW solutions in the regeneration zones of ponds.

## 5. Conclusions

Natural swimming ponds make use of natural solutions based on treatment wetlands (TWs) constituting their regeneration zone. The potential water treatment efficiency is determined by biophysicochemical phenomena associated with the water self-purification process. They take place owing to the functional potential of groups of organisms making up the pond ecosystem, i.e., bacteria, protozoa, fungi, algae, animals, and appropriately selected macrophytes functioning as repository plants, as well as the filtration properties of the substrate. The efficiency of TWs is often supplemented by additional water treatment technologies based on mechanical, mineral, and biological filtration. These processes take place in chambers containing suitably constructed filters with various technological parameters. However, there are few technological water treatment solutions dedicated to natural swimming ponds. These are usually directly transferred from treatment technology for small garden ponds, modified and improved according to the ideas of water technology companies. The diversity of solutions encountered in existing and functioning natural ponds is in most cases the effect of engineering work resulting from practical experience, in

particular the observations and identification of the shortcomings of previous solutions. Therefore, there is a need for an in-depth analysis of the processes taking place in individual devices, particularly physical and biological processes. A flaw observed in many solutions is a lack of modularity allowing the stream of water to easily be switched over into parallel or serial configurations and the easy replacement of individual modules. The modules must be replaceable due to the need for cleaning and regeneration. When this takes place the water circulation is interrupted, which negatively affects the functional efficiency of the entire system.

So that sets of filters can be modified and replaced by non-specialists, the casing must be made of a lightweight material (plastic or aluminium) with appropriate filtration materials for the hydrochemical characteristics of the pond water.

Therefore, there is a need for the modernization and modification of existing water filtration solutions to obtain optimal functional traits, including the following:

- Modularity based on lightweight polymer components (PE or HDPE) and aluminium, which can be easily multiplied as needed;
- The modularity of individual compartments of the filtration chamber itself, including the mechanical, biological, and mineral compartments, as well as a pump compartment with equipment adapted to the quality parameters of the water used to fill the pond and replenish water lost to evaporation;
- The use of a mineral filter with an adsorbent with proven optimal phosphorus sorption efficiency and the possibility of adding another section to the mineral module;
- Functionality for swimming ponds of all sizes, owing to the possibility of modification based on the modularity of the filtration system, without the need for specialist expertise;
- The possibility of selective filtration while any of the cartridges (mechanical, mineral or biological) are idle, without switching off the water circulation in the pond;
- Adaptation of the technological solution for most types of swimming ponds (types 2–5), owing to the possibility of connecting modules into parallel or serial configurations;
- Trouble-free self-assembly of system components based on a starter pack (basic, factory-equipped) and supplementary modules and dedicated accessories chosen for the given conditions;
- A modular system enabling the easy instalment of a deck on the components of the filtration chamber, helping to mask it and optimize the use of the site;
- Easier maintenance, allowing each part of the filtration module to be detached and effectively cleaned, without the need to inactivate the entire system.

**Author Contributions:** Conceptualization, W.W., A.S. and T.S.; methodology, A.S., software, A.S.; validation, W.W., A.S. and T.S.; formal analysis, W.W., A.S. and T.S.; investigation, W.W., A.S., T.S.; resources, W.W., A.S. and T.S.; data curation, W.W., A.S.; writing—original draft preparation, W.W., A.S. and T.S.; writing—review and editing, A.S.; visualization, W.W., A.S. and T.S.; supervision, A.S.; project administration, A.S. All authors have read and agreed to the published version of the manuscript.

**Funding:** This research received no external funding.

**Data Availability Statement:** Not applicable.

**Conflicts of Interest:** The authors declare no conflict of interest.

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
