# Peer review of "Natural Swimming Ponds as an Application of Treatment Wetlands—A Review"

_water, doi:10.3390/w15101878_

Round 1

Reviewer 1 Report

This review summarizes the application of natural swimming ponds as treatment wetlands of swimming water. The structure and functioning of a natural swimming pond as a TW application have been reviewed. The topic of this review is interesting. It may be accepted after major revision.

1.     The abstract is too verbose and the topic is not clear.

2.     The structure of the article is not clear enough, it is recommended to divide it into subheadings.

3.     The introduction is too long, it is recommended to simplify it.

4.     There are too many figures, and I recommend to delete some unimportant ones, such as Figures 4, 8 and 12.

5.     The methods to improve the processing efficiency of natural swimming ponds should be discussed.

6.     The mechanisms of pollution control of natural swimming ponds may be better presented.

Reviewer 2 Report

Dear Authors,

I have analyzed the paper submitted for review and have noted several observations and comments that I believe need to be taken into account.

The paper presents natural swimming ponds that use treatment wetlands (TW) as a swimming water treatment element and proposes a modular and compact filtration system for water treatment by physical and biological methods, made of polymer composites and with replaceable filtration modules.

Observations on the paper:

1. Page 8 - lines 351 - 357 - a series of microphyte plants are listed that you placed in the wet treatment area of a swimming pond in Figure 2. You placed the plants in a certain order.

- Can this order be slightly changed? What would be the criterion?

- What are the characteristics of each individual plant and what role does it play in treating water from swimming ponds?

- It would be advisable to include a table with these plants in which you present their characteristics, the Ellenberg indicator values for the plants in your example, and how this indicator influences the place of the plants in the treatment area, instead of the string on lines 351 – 357.

- Keep the format of the paper template – the lines I refer to do not follow the required format of the paper.

2. Page 8, lines 361 – 364 – does not follow the template of the paper. Please modify.

3. Page 10 – Page 14 - Based on the type of water treatment technology used, present five types of natural swimming ponds with fairly large pictures and smaller schematics for each of these ponds. The pictures could be smaller and under each type of pond, more details are needed about the method used for water purification - description, characteristics, and indicators.

4. Page 14 line 429 - You mentioned the types of filters and stated which would be better for one type of pond or another.

- How did you determine this?

- Have you calculated filtration indicators for ponds using these filters?

- Have you used several types of filters for the same pond and compared the results? Please add these comparisons to a table.

- What water quality parameters did you measure in each pond?

5. A comparison of parameters and indicators is necessary in each of the cases to be able to say that one filter system is better than another. I have not seen definitions of indicators and their calculation methods; I have not seen which water quality parameters are measured. These things must be specified.

6. References [60] and [61] are not active.

In conclusion, I consider that the work cannot be accepted in this form and requires changes to be accepted.

I don't have Comments on the Quality of the English Language.

Round 2

Reviewer 2 Report

Dear Authors,

I have analyzed your paper with the changes made as a result of the reviewers' suggestions.

I consider that the paper can be published in its current form.

Best Regards

Author Response

Dear Reviewer,

Thank you very much for accepting our amendments and positive recommendation of our study for printing in the Water journal.

Sincerely,

College of Authors